# Ubiquitin Ligases in Longevity and Aging Skeletal Muscle

**DOI:** 10.3390/ijms23147602

**Published:** 2022-07-09

**Authors:** David C. Hughes, Leslie M. Baehr, David S. Waddell, Adam P. Sharples, Sue C. Bodine

**Affiliations:** 1Division of Endocrinology and Metabolism, Department of Internal Medicine, Carver College of Medicine, University of Iowa, Iowa City, IA 52242, USA; leslie-baehr@uiowa.edu (L.M.B.); sue-bodine@uiowa.edu (S.C.B.); 2Department of Biology, University of North Florida, Jacksonville, FL 32224, USA; d.s.waddell@unf.edu; 3Institute for Physical Performance, Norwegian School of Sport Sciences (NiH), 0863 Oslo, Norway; adams@nih.no

**Keywords:** protein degradation, proteostasis, E3 ubiquitin ligase, healthspan, sarcopenia

## Abstract

The development and prevalence of diseases associated with aging presents a global health burden on society. One hallmark of aging is the loss of proteostasis which is caused in part by alterations to the ubiquitin–proteasome system (UPS) and lysosome–autophagy system leading to impaired function and maintenance of mass in tissues such as skeletal muscle. In the instance of skeletal muscle, the impairment of function occurs early in the aging process and is dependent on proteostatic mechanisms. The UPS plays a pivotal role in degradation of misfolded and aggregated proteins. For the purpose of this review, we will discuss the role of the UPS system in the context of age-related loss of muscle mass and function. We highlight the significant role that E3 ubiquitin ligases play in the turnover of key components (e.g., mitochondria and neuromuscular junction) essential to skeletal muscle function and the influence of aging. In addition, we will briefly discuss the contribution of the UPS system to lifespan. By understanding the UPS system as part of the proteostasis network in age-related diseases and disorders such as sarcopenia, new discoveries can be made and new interventions can be developed which will preserve muscle function and maintain quality of life with advancing age.

## 1. Introduction

As human life expectancy continues to increase, so does the prevalence of diseases associated with aging and thus factors that contribute towards improved quality of life in older age such as physical activity and cognitive function gain in importance [1,2,3]. Through understanding the physiological mechanisms that contribute to the process of aging, strategies that improve health into later life will reduce the burden of age-related diseases on society [1]. Age-related phenotypes range from a decrease in bone mass (osteopenia), cartilage, and muscle mass (sarcopenia), to frailty and multimorbidity which progress over time and differ in rates between individuals [1]. Examples of the reported hallmarks of aging are mitochondrial dysfunction, cellular senescence, and deregulated nutrient sensing [1,4]. Loss of proteostasis is another hallmark of aging as the ability to preserve protein homeostasis over the course of the lifespan becomes gradually compromised in cells, tissues, and organs [1,4,5] (Figure 1). In age-related diseases (e.g., Parkinson’s disease, Alzheimer’s disease) and aging, damaged and misfolded proteins accumulate due to the progressive decline in the ability of cells to properly maintain the proteome [1,6,7]. In age-related impairment of muscle function and loss of mass, the proteostasis network has been observed to be compromised as evidenced by an imbalance between protein synthesis and degradation [7,8,9,10,11,12,13].

Under basal conditions, the proteostasis network can rapidly detect and rectify any alterations in the proteome and restore protein homeostasis. The ubiquitin–proteasome system (UPS) and the lysosome–autophagy system (LAS) are key in maintaining proteostasis and aid in deciding the fate of unfolded proteins through either non-proteolytic or proteolytic signals [14] (Figure 1). Specifically, the UPS contributes to the proteostasis network by utilizing ubiquitination, a process that involves covalently linking a small protein called ubiquitin to substrate proteins via a series of reactions involving ubiquitin–activating enzymes (E1), ubiquitin–conjugating enzymes (E2), and ubiquitin–protein ligases (E3; ~600 in the human genome) [15,16] (Figure 2). Ubiquitin, a 76-amino acid protein, is most commonly attached through the formation of an isopeptide bond with a lysine residue on the target protein. Furthermore, ubiquitin itself has seven lysine residues (K6, K11, K27, K29, K33, K48, and K63) within its amino acid sequence, which allows for the formation of different types of ubiquitin chains on target proteins (Figure 2). Thus, depending on the number of ubiquitin molecules added, the number of lysines on the protein tagged with ubiquitin, and the type of linkage of the ubiquitin chain, a substrate can be monoubiquitinated, multi-ubiquitinated, or polyubiquitinated [17]. Moreover, the type of ubiquitin linkage and their topology can direct substrate proteins towards either proteolytic or non-proteolytic fates [18]. For example, K48 and K29 chains have been suggested to directly target proteins to the 26S proteasome for degradation, while K63 chains can lead to alterations in signaling and endocytosis [15,17,18,19] (Figure 2). Chaperones also play a key part in conformational changes of proteins throughout the lifespan, from promoting proper folding, complex assembly, and transport across membranes, to targeting for degradation. The activity of chaperones has been shown to decline with age [5,14,20]. Previously published reviews and perspectives have highlighted in detail the interplay of key components in the proteostasis network which are involved in protein synthesis, folding, disaggregation, and degradation [5,14,21,22].

As the UPS is a complex system which contributes to the fine-tuning of the proteostasis network, the E3 ubiquitin ligases best exemplify this complex system through the type of E3 family subgroups [18]. In mammals, two major classes of E3 ligases are prominent: HECT (homologous to the E6AP carboxyl terminus) and RING (really interesting new gene) domains [15,18,23]. These two domains have differing actions for substrate protein ubiquitination, where the HECT E3 ligase has catalytic activity to attach ubiquitin to the substrate protein and the RING E3 ligase functions as a scaffold to bring the E2 and substrate protein together. The Cullin-RING E3 ligase (CRL) superfamily is a multi-subunit complex, which requires a Cullin protein (one of 8 members) to act as a scaffold protein for substrate adaptors and a RING-domain-containing protein (Rbx1 or Rbx2) that associates with the Cullin protein and recruits the ubiquitin charged E2s (reviewed in [24]). The array of E3 ligases in the CRL family and assembly combinations allow for broad substrate protein specificity to be achieved but also add to the complexity when attempting to identify and target E3 ubiquitin ligases for interventions [24,25]. To note, Cullin-RING ligase activity has been implicated in skeletal muscle homeostasis and function in mammalian models [24,26]. Fbxo32 (muscle atrophy F-box protein, MAFbx) and Fbxo30 (muscle ubiquitin ligase of SCF complex in atrophy-1, MUSA1) are examples of E3 ligases that belong to the CRL family and have been observed to be transcriptionally active in skeletal muscle wasting [27,28,29].

## 2. Ubiquitin Proteasome System and Longevity

Given the importance of the UPS in the proteostasis network and the role that ubiquitin has as a signal for post-translation modifications and substrate selection [5,6,30], there have been numerous studies using model organisms to investigate various components of the UPS in aging and longevity [31,32,33,34,35,36,37] (Table 1). As proteasome genes are evolutionarily conserved, non-vertebrate models, such as *Caenorhabitis elegans (C. elegans)*, have been used to investigate major players in the UPS system and findings have been extrapolated to human cells [38]. A plethora of literature has been reported on the roles of Insulin/IGF, mTOR (mammalian/mechanistic target of rapamycin), the Sirtuins and Forkhead Box O (FOXO) transcription factors in the process of lifespan and longevity (reviewed in [3,39,40,41,42]). Briefly, reductions in Insulin/IGF and mTOR signaling are observed to modulate lifespan and increase healthspan in various model organisms [43,44,45,46,47,48]. FOXO transcription factors act as key regulators of longevity downstream of insulin/IGF signaling (reviewed in [39,49]). The FOXO transcription factors display an ancestral and conserved role in the processes of metabolism, growth factor signaling, stress resistance and proteostasis [39,50]. Further, the FOXO transcription factors are involved in targeting and activating muscle-specific E3 ubiquitin ligases including muscle-specific RING finger protein 1 (MuRF1) [51], MAFbx [52], specific of muscle atrophy and regulated by transcription (SMART) [53], MUSA1 [28] along with other lysosome–autophagy related proteins involved in protein degradation [54]. Numerous ubiquitin ligases have been shown to regulate the Insulin/IGF pathways and other pathways involved in dietary restriction interventions [31,32,37,55,56].

The E3 ubiquitin ligase WW Domain-Containing Protein 1 (WWP-1) was identified by Carrano and colleagues [32] in lifespan extension under ad libitum feeding and dietary restriction conditions using the *C. elegans* model. The overexpression of a mutant form of WWP-1 which lacks ubiquitin ligase activity was observed to suppress the increased lifespan of dietary restricted *C. elegans*. This observation indicates that the activity of the ubiquitin ligase is essential for the longevity response under dietary restriction conditions. A follow up study by Carrano and colleagues extended their previous work by identifying Kruppel-like factor-1 as a substrate for monoubiquitination by WWP-1 [55]. Notably, from these two studies [32,55], was the observation that lifespan extension through WWP-1 were independent of the FOXO transcription factor family member, DAF-16. DAF-16 is important in lifespan regulation, as its activation leads to DAF-16 nuclear accumulation and controls the transactivation/repression of numerous targets genes critical for stress response and metabolism [57,58]. Notably, DAF-16 has been demonstrated to influence longevity through the modulation of mTOR signaling, AMPK and JNK pathways as it can integrate signals from these pathways via its translocation from the cytoplasm to nucleus via Akt or JNK phosphorylation [43,58,59,60]. Many factors have been suggested to influence the regulation of DAF-16 as a central regulator of the stress response to aging [57]. The E3 ubiquitin ligase, Regulation of Longevity by E3 (RLE-1) has previously been identified in the modulation of aging by the ubiquitination of DAF-16 [31]. Disruption of RLE-1 in *C. elegans* leads to extended lifespan, through elevations in DAF-16 protein and sustained localization of DAF-16 at the nucleus. It was observed by Li and Colleagues [31] that RLE-1 promotes DAF-16 ubiquitination, yet little is known about how the Insulin/IGF pathway responds to RLE-1 mediated ubiquitination of DAF-16. In the *C. elegans* model, the identification of the ubiquitin ligases WWP-1 and RLE-1 provide evidence for the important role of the UPS in longevity, yet future research is required in mammalian models.

Another E3 ubiquitin ligase has been suggested to promote lifespan via pathways distinct from dietary restriction and insulin/IGF-I signaling [61]. In response to hypoxia, the von Hippel–Landau tumor suppressor homolog VHL-1 has been observed to negatively regulate the hypoxic response by targeting HIF-1 for ubiquitination and subsequent degradation [61]. VHL-1 is a member of the Cullin-RING E3 ubiquitin ligases. Mehta and colleagues [61] observed increase lifespan and enhanced resistance to proteotoxic stress with the loss of VHL-1. Further, the authors speculated that DAF-16 target genes may be shared with HIF-1 [62], and thus regulation of HIF-1 via VHL-1 may contribute to the modulation of stress resistance and lifespan. Evidence suggests that VHL-1 may modulate Akt activity as VHL-1 has been observed in cancer cells to suppress Akt kinase activity [63]. Thus VHL-1 is another E3 ubiquitin ligase which may intersect with the insulin/IGF-1 signaling pathway, although the role of VHL-1 appears to be critical in response to hypoxic conditions.

Evidence has alluded to a group of E3 ubiquitin ligases regulating the activity of AMPK activation (a key regulator of cellular energy homeostasis) and lifespan [37,64]. The glucose-induced degradation deficient (GID)-protein complex is evolutionarily highly conserved and targets key enzymes in gluconeogenesis for 26S proteasomal degradation [64,65]. The complex consists of individual subunits (e.g., Gid1, Gid4), with the RMND5A protein subunit being critical for ubiquitin ligase activity of the complex [64]. In the study by Liu et al. [37], the authors observed decreased AMPK ubiquitination in RMND5A knockout cells (NIH-3T3), which corresponded to the cells displaying increased AMPK activity, autophagic flux, and low energy status. The knockout of RMND5A in *C. elegans* resulted in increased lifespan and additional AMPK activation within this model. The observations by Liu and colleagues [37] highlight the potential interplay between E3 ubiquitin ligases and the AMPK-mTOR axis that is extensively studied in organismal lifespan. Other E3 ubiquitin ligases (e.g., MuRF1) have been observed to participate in the regulation of metabolic processes such as amino acid availability [66,67,68]. Overall, the importance of the GID complex in mammalian models towards AMPK regulation and lifespan control is warranted.

The quality control E3 ubiquitin ligase CHIP (carboxyl terminus of Hsp70-interacting protein) has been observed to regulate insulin receptor turnover through monoubiquitination, where CHIP deficiency leads to reduced lifespan in *C. elegans* and Drosophila [56]. Interestingly, the study by Tawo et al. [56] observed that under proteotoxic stress conditions and aging, the role of CHIP is predominately at the disposal of misfolded proteins and thus insulin receptor degradation is reduced. It was noted by Tawo and colleagues [56] that overexpression of CHIP was detrimental towards lifespan in flies. In whole body knockout mice, CHIP deficiency significantly reduced mammalian lifespan and was accompanied by reduced proteasome activity in muscle and liver tissues [36]. Interestingly, longevity pathways (e.g., FOXOs, Sirtuins) were not changed within CHIP-deficient mice. The examples of WWP-1, RLE-1 and CHIP highlight the complex and competitive balance that the UPS system and proteasomal degradation play in regulating the proteome and cellular response to stress during aging (Table 1).

With regards to the impact of the UPS in skeletal muscle on lifespan, a study by Kitajima and colleagues [35] utilized a muscle-specific conditional knockout of the 26S proteasomal regulatory subunit 6B homolog (Rpt3) proteasomal gene. Rpt3 is an essential subunit of the 26S proteasome core (Figure 2) and required for degradation of substrates selected to the proteasome. In Rpt3 knockout muscles, the authors observed significant loss of muscle mass and fiber size, dysregulation of the UPS system and notably a shortened lifespan verses wild type mice. In CHIP deficient mice not only was lifespan significantly reduced but age-related loss of muscle was observed compared to age-matched controls, as assessed in the gastrocnemius and quadriceps muscles [36]. Further, 26S proteasome activity was significantly decreased in skeletal muscle lysates from CHIP-deficient mice compared to wild type controls. These observations highlight the critical role of the UPS in skeletal muscle maintenance and the effect of a dysregulated skeletal muscle UPS on lifespan.

In aging and healthspan, skeletal muscle is an important tissue for the ability to maintain quality of life through performing daily activities and prolonging overall independence. A common mechanism which is central to the turnover of skeletal muscle and longevity is mTORC1 signaling, where in skeletal muscle the nutrient-sensing mTORC1 pathway promotes muscle growth yet is pivotal in the age-related loss of muscle mass [69,70,71]. Indeed, sustained mTORC1 hyperactivation is observed to induce the molecular signatures of age-related muscle loss [70,71]. Recent evidence suggests that mTORC1 might also play an important role in the activation of the UPS system and overall muscle proteostasis [72]. The mTORC1 pathway is an example of the cellular and molecular mechanisms which contribute to the age-related loss of muscle mass but other pathways have been implicated in longevity, which help modulate skeletal muscle maintenance [3].

**Table 1 ijms-23-07602-t001:** Summary of E3 ubiquitin ligases identified in the regulation of lifespan.

E3 Ligase	Model	Expression	Lifespan	Reference
WWP-1	*C. elegans*	overexpression	Increased	Carrano et al. [32]
	*C. elegans*	overexpression	Increased	Carrano et al. [55]
RLE-1	*C. elegans*	Knockout	Increased	Li et al. [31]
CHIP	Mouse	knockout	Decreased	Min et al. [36]
	*C. elegans*	Knockout	Decreased	Tawo et al. [56]
	Drosophila	Knockout	Decreased	Tawo et al. [56]
Parkin	*C. elegans*	Overexpression	Increased	Rana et al. [73]
RPT3	Mouse	Knockout (muscle-specific)	Decreased	Kitajima et al. [35]
GID Complex (RMND5A)	*C. elegans*	knockout	Increased	Liu et al. [37]
VHL-1	*C.elegans*	Knockout	Increased	Metha et al. [61]

## 3. Brief Overview of Aging Skeletal Muscle

The etiology of age-related loss of muscle mass and strength, which can be referred to as sarcopenia and dynapenia, respectively [13,74,75], has been extensively studied in recent decades. This area of research is increasingly pertinent as the proportion of older individuals increases in society and results in elevated incidences of falls and fall-related injuries [76,77,78]. The onset of age-related muscle mass loss occurs between 30–40 years of age [79,80], and while the development of sarcopenia is complex and multifactorial, there have been numerous avenues of investigation focused on understanding the underlying cellular and molecular mechanisms [13]. The etiology of sarcopenia has been investigated in relation to inflammation, mitochondria, motor unit function, contractile fiber size, and force transmission (detailed review in [13,81,82,83,84,85]). The contractile activity of skeletal muscle is controlled by a neuronal component through the neuromuscular junction (NMJ), while the aging process is believed to contribute to a decline in NMJ transmission and force generation [86]. Three elements comprise the NMJ: (1) the pre-synaptic zone located in the motor nerve terminal, (2) the intra-synaptic basal lamina and, (3) the post-synaptic acetylcholine receptors (AChRs) located along the motor end plate of the muscle. Binding of acetylcholine to the AChRs is critical for skeletal muscle contraction [87,88,89]. In human and rodent muscle, dysfunction in the NMJ has been highlighted in aged skeletal muscle through either endplate fragmentation and/or upregulation of markers involved in muscle denervation [90,91,92,93,94,95]. The overall life cycle of AChRs decline with age [86,96] and recent studies have suggested the possible involvement of the UPS and autophagy systems in AChR turnover [97,98,99] (Figure 3).

In addition to NMJ dysfunction, impaired mitochondrial function is evident in aged skeletal muscle as reported through alterations in mitochondrial respiration, content, and turnover [100,101,102,103,104,105,106]. The impairment of mitochondrial function results in reduced ATP production and an inability to meet the energetic demands of protein turnover, metabolism, and cellular repair [107,108,109]. Mitochondrial fusion, fission, and autophagy (mitophagy) have been studied as key processes involved in mitochondrial quality control [104,110]. Fusion and fission allow for the regulation of the mitochondrial network size, whereas mitophagy is key in the identification and removal of damaged mitochondria by formation of autophagosomes and degradation via the LAS [111]. Moreover, numerous studies have observed functional and expression changes of key proteins involved in the clearance of damaged mitochondria in aged skeletal muscle [112,113,114]. The examples of the dysregulation of the NMJ and mitochondrial network in aged skeletal muscle highlight the importance of the proteostasis network. To note, in rodent models, NMJ and mitochondrial dysfunction have been observed to precede the onset of the sarcopenic phenotype [115,116]. The focus of this review going forward is to highlight the current literature and recent developments pertaining to the involvement of E3 ubiquitin ligases and the UPS in aged skeletal muscle.

## 4. Ubiquitin Ligases in Sarcopenic Skeletal Muscle

Commonly E3 ubiquitin ligases have been identified and characterized in skeletal muscle atrophy conditions, ranging from disuse to metabolic stress [27,28,52,117,118,119,120,121,122,123,124,125]. In the instance of aged skeletal muscle, E3 ubiquitin ligases have received little investigation, with only a handful of ubiquitin ligases being studied [114,126,127,128,129]. Early observations in the published literature have focused on ubiquitin levels, proteasome content, and activity in various rodent aging models [130,131,132,133,134]. In a study by Cai et al. [132], the authors observed that ubiquitin levels were increased in the EDL muscle of 24-month-old rats and in quadricep muscle biopsies of 70–79 year old human patients compared to adult (20–29 years) skeletal muscle. The authors also noted that the increased level of ubiquitin was specific to fast twitch muscles, as highlighted by a comparison of the rat extensor digitorum longus (EDL) versus soleus muscle. However, Chen et al. did observe elevated total ubiquitin, as determined via immunoblotting, in the slower soleus muscle of middle-aged rats versus young (4-months) and adult (8-months) skeletal muscle [135]. A subsequent study by Clavel and colleagues also identified increased ubiquitin levels in aged tibialis anterior (TA) muscles of Sprague–Dawley rats (24 months old) [133]. Under basal conditions, our lab has recently observed no differences for total ubiquitin levels in the soleus and TA muscle of 29-month-old male Fischer 344-Brown Norway (F344BN) rats compared to 9-month-old adult controls [124]. It is possible that the increased presence of ubiquitin as a signal might be age-sensitive and muscle-specific. However, changes in total ubiquitin levels with aging appear to be inconclusive and subsequent research is required to delve deeper into the role of ubiquitin and this post-translational modification in aged skeletal muscle.

In terms of proteasome activity and content in aged skeletal muscle, Altun and colleagues [12] reported observations in hindlimb muscles from 30-month-old aged rats showing 2–3-fold higher levels of 26S proteasomes compared to the younger 4-month-old adult animals. Further, the authors noted significantly elevated levels of proteasome activity in the 30-month-old skeletal muscle, as measured by capase-like and chymotrypsin-like assays. Similar results were observed by Hepple and colleagues in the plantaris muscle of 30–35-month-old aged rats, where proteasome activity was reported to be increased [131]. In a study by Wallace and colleagues [136], the quadriceps muscles of 26-month-old mice displayed increased proteasome activity for the 20S (β1, β2, and β5 subunits) and 26S (β1 and β5 subunits) proteasomes compared to mice at 16-months of age. In our lab, we have observed muscle-specific changes in proteasome activity with aging [124,126,137]. We observed in the TA muscle of 29-month-old F344BN rats increased 20S and 26S proteasomal activity, whereas there was no difference observed in the soleus muscle compared to 9-month-old F344BN adult controls under basal conditions [124]. However, we have observed decreased basal proteosome activity (20S and 26S β5) in the gastrocnemius complex of 24-month-old mice and reduced 26S (β1) and 20S (β2 and β5) proteasome activity in the TA muscle of 30-month-old F344BN rats [126,137]. To note, studies have observed proteasome activity to be unchanged or decreased between adult and old skeletal muscles of animals [124,133,138,139,140], which may be explained by the differences in skeletal muscles analyzed, the age of the animals, and the type of proteasome activity assay being performed [12,124,131,133,139,140]. In addition, a factor to be considered is the possible disparity between proteasome activity and alterations in subunit composition (e.g., reduced proteasome complex protein content) [130,134]. Recent studies have observed shortened lifespan in mouse models where reduced proteasome activity is induced through muscle specific deletion of the proteasome gene, Rpt3 (also known as Psmc4) [35,141]. Overall, numerous studies have suggested that aged skeletal muscle exhibits elevations in proteasome activity and ubiquitination as a result of dysfunctions in various cellular processes involved in protein turnover and homeostasis, such as synthesis and misfolding.

Limited studies have been performed that investigate the roles of specific E3 ubiquitin ligases in sarcopenic muscle. Since their discovery in 2001, MuRF1 and MAFbx/Atrogin-1 have been extensively studied under muscle wasting conditions [27,117,118,142]. However, studies in sarcopenic muscle have revealed mixed results with regards to the involvement of MuRF1 and MAFbx in aging (Table 2). A study by Clavel and colleagues [133] observed mRNA expression for MuRF1 and MAFbx to be significantly elevated in the TA muscle of 24-month-old Sprague–Dawley rats compared to 5-month-old adult controls. In comparison, studies have reported MuRF1 and MAFbx to be downregulated or unchanged in aged rodent skeletal muscle [124,142,143,144,145]. The notable differences in these studies compared to the Clavel et al. study is the selection of hindlimb muscle (i.e., TA versus gastrocnemius (GA)) and the age of the rodents (24 vs. 30-month-old). Indeed, recently we have observed no differences for basal MuRF1 and MAFbx mRNA expression in soleus, TA, or GA muscles of adult and old (29-month-old) F344BN rats, yet aging did alter the expression response to periods of disuse atrophy and muscle regrowth [124,142]. There is conflicting evidence regarding changes in MuRF1 and MAFbx expression in sarcopenic muscle from rodents compared to humans (Table 2). In physically inactive frail older women, Drummond and colleagues observed significant reductions in MuRF1 and MAFbx expression in muscle biopsies from the vastus lateralis [146], whereas Raue and colleagues reported a significant elevation in MuRF1 expression and a slight increase in MAFbx expression at rest in older women (85 ± 1 years) compared to young women (21–25 years) [147]. Other previous studies have observed no differences in the basal expression levels of MuRF1 and MAFbx in young and older individuals using human muscle biopsy samples [148]. It is important to note that observations on MuRF1 and MAFbx expression and protein levels in skeletal muscle are made from whole muscle homogenates and thus changes could be occurring at the fiber level and not detected. Overall, the levels of MuRF1 and MAFbx and their association with the loss of age-related muscle mass and function remain contentious.

From an experimental model perspective using genetic manipulation, our lab has studied the effect of MuRF1 deletion on aging skeletal muscle [126]. The deletion of MuRF1 in 24-month-old mice led to significant muscle mass and fiber CSA size sparing compared to aged matched wild type littermates. The sparing of muscle mass in these 24-month-old MuRF1 KO mice was accompanied by elevated proteasome activity and reductions in markers of ER stress (BiP, PDI, CHOP protein). Most notably, even though sparing of muscle mass occurred, measurements of contractile force output (in situ) displayed significant reduction in 24-month-old KO muscle verses age-matched wild-type controls. One proposed mechanism for the reduced force output in MuRF1 KO muscles may center on neural mechanisms and neuromuscular junction (NMJ) stability due to the reported involvement of MuRF1 in acetylcholine receptor turnover [97,98,99]. We have observed MuRF1 KO muscles to display a reduced abundance of HDAC4 with denervation [119]. HDAC4 has been observed to play a central role in the regulation of synaptic genes, as the onset of denervation leads to increased HDAC4 abundance and localization from the cytoplasm to the nucleus [149,150]. Recent evidence alludes to HDAC4 activity being controlled by mTORC1, where hyperactivation of mTORC1 leads to neuromuscular endplate degeneration and impacts synaptic remodeling [151]. The role of HDAC4 and MuRF1 under innervation and denervation cycles in skeletal muscle remains to be explored in aged tissue.

Although the location of MuRF1 and its substrates in skeletal muscle are suggested to be at the contractile apparatus (e.g., titin) [15,152,153,154], there are recent studies identifying the presence of MuRF1 at the NMJ (Figure 3) [97,98]. A study by Rudolf and colleagues [98] highlighted enrichment of MuRF1 at the NMJ in mouse EDL muscles under basal conditions and the tibialis anterior muscle during denervation-induced atrophy. Through the use of affinity precipitation and in vivo imaging techniques, the authors observed MuRF1 interacting with endocytic structures containing acetylcholine receptors and Bif-1 (Bax-interacting factor 1, also known as SH3GLB1 or Endophilin B1) that forms a complex with Beclin 1 and is involved in autophagosome formation [155]. The deletion of MuRF1 in mouse quadriceps muscle also led to a reduction in AChR degradation and subsequent muscle denervation [98]. Recently, our lab found that MuRF1 overexpression in adult mouse muscle is sufficient to induce skeletal muscle atrophy and leads to an increase in expression of genes (AChRα, Gadd45a, MuSK) associated with NMJ instability [156]. Other research groups have similarly highlighted the importance of MuRF1 in AChR degradation, through identifying a potential pathway (Gα_i2_-HDAC4-Myogenin-MuRF1) for motor denervation, NMJ transmission and muscle force generation which is regulated by the sympathetic nervous system (SNS) [99]. Further, myogenin, MuRF1, and acetylcholine receptor subunit α1 (AChRα) have been observed to display reduced DNA methylation in an atrophy model using tetrodotoxin injections which silence the neural input into the hindlimb muscles [157]. Overall, future studies to investigate the effect of MuRF1 over the lifespan on AChR turnover and force production in aging skeletal muscle are warranted. Given the large number of studies detailing the importance of NMJ maintenance/function, and evidence of denervation in sarcopenic muscle [13,71,91,92,93,115,158,159,160], understanding NMJ turnover and the possible molecular targets involved over the course of the lifespan would enhance our knowledge in the understanding for the development of sarcopenia [13].

E3 ubiquitin ligases have continuously been highlighted for their role in the turnover of sarcomeric proteins and regulation of the contractile apparatus, albeit predominately under conditions of atrophy [121,122,123,161,162]. A recent study by Seo and colleagues [129], explored the role of Mind Bomb-1 (Mib1) in sarcopenic muscle using human muscle biopsies and the generation of myofiber-specific Mib1-deficient mice. Mib1 is an E3 ubiquitin ligase which is reported to regulate notch signaling and may be implicated in tissue maintenance, cell differentiation, and mechanical signals in skeletal muscle [163,164,165]. The authors reported an age-related loss of Mib1 in skeletal muscle from 16 to 30 months compared to 3-month-old mice. Further, aged Mib1 deficient mice displayed muscle atrophy and impaired function by 16 months of age. Mechanistically, the authors attributed the age-related muscle phenotypes in Mib1 deficient mice to an accumulation of alpha-actinin 3 (Actn3), due to a lack of proteasomal degradation mediated by Mib1, culminating in specific atrophy of myosin heavy chain 2b fibers. The overexpression of Actn3 has been observed to be detrimental to skeletal muscle function [166]. Aged rodent models display an accumulation of contractile apparatus proteins that range from the dystrophin-glycoprotein complex (DGC) to z-disk proteins such as alpha-actinin, desmin, and desmuslin [85,167,168,169]. Given that lateral force transmission is compromised in aged rodent skeletal muscle [170,171], the study by Seo et al. [129] highlights the potential role of E3 ubiquitin ligases in sarcomeric turnover with age and the maintenance of skeletal muscle integrity and force production [85].

The HECT (homologus to E6AP carboxy terminus) family member UBR4 (~600 kda), has recently been identified to be involved in protein quality control and muscle mass regulation in aging [128,172]. In aged muscle, UBR4 protein is elevated and increases the proteolytic activity of the proteasome, while UBR4 expression was found to increase in response to proteostasis stressors [128]. In one study by Hunt et al. [128], muscle-specific loss of UBR4 was found to rescue age-associated muscle mass and myofiber atrophy, but there was an impairment in normalized muscle force. Interestingly, using *Drosophila*, the authors found that genes which negatively regulated myofiber size were critical in protein quality control, muscle function, and organismal lifespan. In an earlier study by Hunt and colleagues, the authors identified a role for UBR4 in myofiber hypertrophy through RNAi screening in Drosophila, mice, and the C2C12 myoblast cell line [172]. The authors reported fiber type specific increases in tibialis anterior CSA with the loss of UBR4 and in mice demonstrated that UBR4 interacted with proteins involved with the UPS (e.g., UBE2B, ASB8, and KLHL30) and those with nuclear functions (e.g., PCNA, HAT1, and RBBP4). These studies highlight the importance of protein quality control in muscle mass regulation via ubiquitin ligases and the need for therapeutic interventions in order to balance the need of a combined approach towards targeting both muscle mass and strength together.

As previously mentioned, an important element in aging skeletal muscle and the proposed etiology of sarcopenia is mitochondrial function and turnover [13]. There is a plethora of evidence for the role of mitochondrial dysfunction in the development of sarcopenia [103,173,174,175]. Dysfunctional mitochondria are removed by mitophagy, which is regulated by the proteins PINK1 and Parkin [110,176]. Parkin is an E3 ubiquitin ligase, encoded by the *Park2* gene and is activated by phosphatase and tensin homologue-induced kinase 1 (PINK1) which leads to ubiquitination of mitochondrial outer membranes proteins such as TOMM20 and VDAC [177,178,179,180]. Subsequently, this ubiquitination signal triggers the activation of the UPS system and autophagic processes needed for removing dysfunctional mitochondria [181,182]. In aging skeletal muscle, there is limited direct evidence for the process of mitophagy, but recent studies have observed increases in mitophagy-related proteins with aging [102,112,114,183]. Further, multiple studies detail the accumulation of mitochondrial damage [184] and impairment [100,101,175,185,186], suggesting that mitochondrial clearance is compromised with age.

In a study by Chen and colleagues, the authors reported increased Parkin protein levels in whole muscle lysates from 18-month-old mice but significantly reduced amounts of Parkin localized in the mitochondria [114]. In contrast, a study by Yeo et al. utilizing 24-month-old mice reported a significant increase in mitochondrial Parkin [102]. At the transcript level, Drummond and colleagues noted significant reductions in mitophagy related genes in muscle biopsies from elderly (83 ± 2 years) frail women compared to elderly active (77 ± 2 years) controls [146]. Animal and in vitro models using genetic manipulation of Parkin in skeletal muscle have provided insight into the importance of this E3 ubiquitin ligase in mitochondrial turnover, muscle function, and aging [114,127,187,188,189,190]. A study by Gouspillou and colleagues [189] utilized whole body Parkin deletion in mice (Parkin -/-) and reported significant decreases in skeletal muscle mitochondrial function (respiration, uncoupling etc.) as well as reduced specific force. Similar observations have been made in C2C12 myotubes, where the introduction of Parkin siRNA lead to impaired mitochondrial function and subsequently the onset of myotube atrophy [187]. In addition, Gouspillou et al. observed an increase in pro-fission protein DRP1 which suggests an increase in mitochondrial fragmentation. Interestingly, Peker and colleagues [187] also assessed DRP1 in Parkin -/- muscles and noted a significant decline in protein levels. Notably in other tissues, DRP1 is suggested to be a substrate that is targeted by Parkin for UPS degradation [191], yet the substrates for Parkin in skeletal muscle remain to be determined.

Genetic manipulation of Parkin in aging skeletal muscle has provided insight into the role of this E3 ubiquitin ligase as a potential therapeutic target for sarcopenia. A recent study by Leduc-Gaudet and colleagues [127] overexpressed Parkin (via adeno-associated virus injection) in 3-month-old and 18-month-old mice for a period of 4 months. The authors reported a significant increase in TA and GA muscle mass and fiber size for young 7-month-old adult and aged 22-month-old mice. In addition, the authors reported significant increases in force output of aged skeletal muscle with Parkin overexpression compared to controls. The phenotypic changes with Parkin overexpression were accompanied by increased mitochondrial content, enzymatic activity, and reduced oxidative stress. Moreover, two recent studies overexpressed Parkin in Drosophila [73,192] and both studies observed extensions in lifespan. The study by Si and colleagues [192] reported suppression of mitochondrial ubiquitination and the restoration of ATP levels through an autophagy dependent manner (Atg1), as assessed in indirect flight muscles. An earlier study by Rana et al. [73] reported that alongside an increase in mitochondrial activity (citrate synthase, complex I and II), there were smaller mitochondria present in young and aged indirect flight muscles when Parkin was overexpressed. The authors attributed this observation to a reduction in mitofusion levels, a more segregated mitochondrial network and improved clearance of damaged mitochondria. Future studies are required to expand our understanding of Parkin’s role in mitochondrial quality control in animal and human skeletal muscle plasticity and aging [104,110,173].

Although not directly investigated in aging skeletal muscle, numerous E3 ubiquitin ligases have been researched for their possible role in muscle mass regulation and function [120,172,193,194,195]. Recently, our labs identified a HECT family member protein named UBR5 (~300 kda) that is capable of regulating muscle mass size and is responsive to mechanical loading [120,193,196,197]. Moreover, we observed the involvement of UBR5 in multiple experimental models of muscle hypertrophy and regrowth in both rodent and human models [120]. Most recently, we observed UBR5 knockdown in mouse TA muscle leads to significant skeletal muscle atrophy which coincided with chronic hyperactivation of mTORC1/S6K1/S6 signaling and alterations in ERK1/2 and p90RSK signaling [193]. There is evidence that UBR5 phosphorylation (serine 1543) is increased following mechanical load-induced muscle hypertrophy by 88% even in the presence of rapamycin [198]. Future research is needed to investigate if the muscle atrophy induced by UBR5 knockdown occurs via mTORC1-dependent or independent mechanisms. A study by Shi and colleagues [194] identified the role of Fbxo40 in the ubiquitination of insulin receptor substrate-1 (IRS-1) and observed muscle hypertrophy in Fbxo40 knockout mice. Further, the authors used C2C12 myotubes to show that the Skp-1-cullin1-Rbx-1 complex was responsible for the proteasomal degradation of IRS-1. Another E3 ubiquitin ligase, mitsugumin 53 (MG53; also known as TRIM72) has been reported to ubiquitinate IRS-1 and modulate IGF-induced IRS-1 activation [199,200,201]. Additionally, Lee and colleagues [200] observed enhanced myogenesis in satellite cells from MG53 null mice and MG53 promoter activity was induced by Akt and MyoD. Notably, Song et al. discovered that MG53 overexpression induced insulin resistance and metabolic syndrome in transgenic mice compared to wild type controls [201]. On the contrary, a recent study by Philouze and colleagues [202] provided evidence that MG53 is not a critical regulator of insulin signaling using in vivo and in vitro models, as MG53 gene deficiency was not protective against diet-induced obesity or glucose intolerance. However, the example of MG53, which has also been implicated in the molecular response to muscle injury repair [203,204,205,206], does highlight the various roles that E3 ubiquitin ligases can play in skeletal muscle plasticity and metabolism. In the instance of Fbox40, MG53, and UBR5, these E3 ubiquitin ligases show evidence of an interaction with anabolic pathways in skeletal muscle, the significance of which warrants further investigation as it may have implications in sarcopenia and longevity [3].

## 5. Effects of Exercise and Dietary Interventions on the Ubiquitin Proteasome System in Aged Skeletal Muscle

Numerous intervention strategies to promote healthy aging and muscle mass maintenance have ranged from dietary restriction to different modes of exercise, where some of the beneficial effects of these interventions aid in the maintenance of the proteostasis network [3,109,207,208,209]. Caloric restriction (CR) has been observed in various mammalian models to enhance lifespan and reduce the incidents of pathophysiology manifestations in aging (Reviewed in [3,210,211]). Caloric restriction is typically indicative of a 20–40% sustained reduction in daily energy intake and avoids invoking malnutrition [211]. In skeletal muscle, several studies have assessed the effect of caloric restriction on the proteasome system in the context of aging [12,131,135,140]. A study by Hepple and colleagues [131] observed no differences in MuRF1 and MAFbx mRNA expression in the plantaris muscle of calorie restricted aged rats compared to ad libitum aged matched controls. The authors did note a delayed increase in proteasome activity in skeletal muscle following CR, which was proportional to the lifespan extension observed in calorie restricted animals. In contrast, Altun and colleagues [12] observed a significant decrease in proteasome activity and content in gastrocnemius muscles from 30-month-old rats on a calorie restricted diet compared to ad libitum aged-match controls. In addition, Altun et al. observed no differences in levels of poly- or mono-ubiquitin, but did observe reductions in deubiquitylating enzyme activity (USP5, USP15 and UCH37) in aged animals undergoing CR. Notably, aged skeletal muscle from the CR group was preserved compared to the aged matched control group. A study by Chen and colleagues implemented a 40% CR diet for 14 weeks in 16-month-old Sprague–Dawley rats and observed a significant reduction in ubiquitinated proteins with the CR diet compared to ad libitum controls. An important factor that may influence the differences in the effectiveness of CR on the proteasome system may be the age when the dietary interventions are implemented and overall diet duration. An emerging dietary intervention, the ketogenic diet, has been observed to enhance healthspan and longevity [212,213] in rodent models and preserve skeletal muscle mass and function with age [136]. In the study by Wallace and colleagues [136], the ketogenic diet intervention led to reduced proteasome activity (20S β5 and 26S β5) in quadriceps muscles from 26-month-old mice compared to isocaloric age matched control mice. In addition, the authors noted decreases in protein synthesis, ER stress, and slowed translation initiation which may have contributed to improved proteostasis and mitigated the age-related loss of muscle mass and function.

Several studies have investigated the effect of acute exercise on the UPS in skeletal muscle [148,214,215,216,217]. In young and older adults, Fry and colleagues [215] observed increases in MuRF1 mRNA expression at 3 and 6 h post resistance exercise after muscle biopsies from the quadriceps (vastus lateralis) in both cohorts. Similar observations were reported in quadricep (vastus lateralis) muscle biopsies from young and older participants, where MuRF1 mRNA expression was elevated 2 h after a single bout of resistance exercise [148]. However, increases in MuRF1 and MAFbx following eccentric exercise are usually short lived and both ligases are usually suppressed under growth conditions, even though proteasome activity may increase [218]. Recent evidence has highlighted the global dynamic response of the ubiquitinome following an acute high intensity exercise stimulus [217]. A study by Parker and colleagues [217] assessed diGly tag enrichment in quadricep (vastus lateralis) muscle biopsies from healthy untrained adults at pre-exercise, immediately after exercise, and 2 and 5 h post exercise. The authors reported a rapid clearance of protein ubiquitylation, consistent with proteasome activation, which was returned to pre-exercise levels after 2 h. Further, the authors identified the presence of cross-talk in ubiquitylation and phosphorylation, where 43 proteins displayed phosphorylation and ubiquitylation simultaneously during exercise, which was attributed to the role of cAMP-dependent signaling of the Cullin-RING ubiquitin ligase regulator, NEDD8 [219]. The identification of post-translational modifications adjacent or in close proximity may serve to aid in binding sites for subsequent modifications or alter the protein structure, reducing the possibility of secondary modifications to occur [220]. It should be noted that the identification of 43 proteins was achieved by integrating data sets from the Parker et al. study with a phosphoproteomic data set from a study which used an identical exercise protocol [221]. These data highlight the extensive network of post-translation modifications and role the UPS plays in the dynamic process of clearing potentially damaged proteins in skeletal muscle after exercise.

In young adults, Stefanetti et al. [216] assessed molecular markers of the UPS in skeletal muscle following both an acute single bout of exercise and chronic training (including both resistance and aerobic exercise conditions). The authors observed increases in MAFbx, MuRF1, and Fbxo40 mRNA expression independent of training mode, but notably in trained muscle, these ubiquitin ligases displayed a heightened response to a single bout of aerobic versus resistance exercise. In addition, irrespective of the training stimulus, Stefanetti and colleagues [216] noted an elevation in FOXO1 and FOXO3 mRNA expression in aerobic and resistance-trained groups. In animal models, Cunha and colleagues [214] observed enhanced 26S proteasome activity and MuRF1 mRNA expression in the plantaris muscle of healthy mice after 8 weeks of aerobic exercise training. Similar observations were also reported by the authors under an acute exercise intervention (0, 24, and 48 hrs post exercise) [214]. A study by Ziaaldini and colleagues [222] observed no significant effect of aerobic exercise on the level of ubiquitinated proteins or proteasome subunit alpha (PSMA6) in the quadricep muscles of aged male Wistar rats following a 6-week training program. There is, however, a limited number of published studies on the effect of exercise on the UPS in aged skeletal muscle. Future investigations are required in aging and skeletal muscle to assess the response of the ubiquitinome under various exercise modes and following both acute exercise and chronic training.

## 6. Conclusions

As the loss of proteostasis is a hallmark of aging, exploring the fundamental pathways involved in regulating protein turnover and homeostasis is critical to unraveling the complex etiology of age-related diseases. The UPS plays an integral part in the proteostasis network and numerous model organisms have been utilized to investigate the UPS in lifespan and age-related disease. Emerging evidence through newly identified E3 ubiquitin ligases (e.g., Parkin, UBR4, and Mib1) in aged skeletal muscle has brought attention to the importance of degradation systems in protein quality control across the lifespan and their influence on skeletal muscle function. In certain instances, the loss of E3 ubiquitin ligases can have detrimental effects on muscle homeostasis and function, yet an accumulation in other E3 ubiquitin ligases can be just as deleterious. These observations illustrate the challenges in the development of pharmacological and non-pharmacological interventions towards age-related loss of muscle mass and function. With over 600 E3 ubiquitin ligases in the human genome, many remain to be investigated to determine their effects on skeletal muscle plasticity, aging, and longevity.

## Figures and Tables

**Figure 1 ijms-23-07602-f001:**
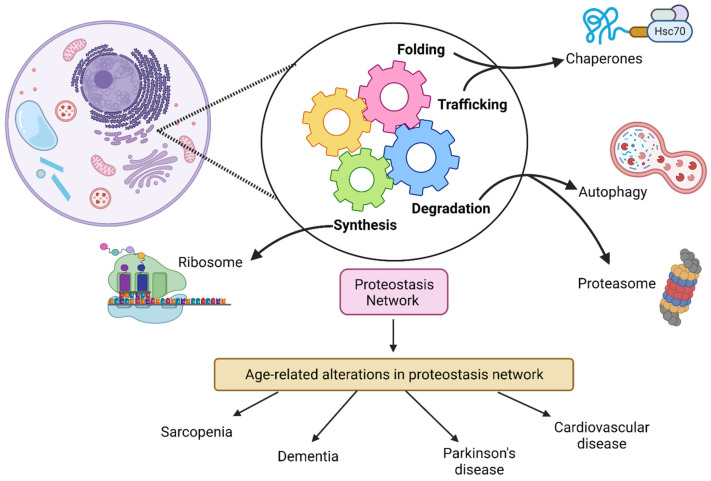
**Overview of the proteostasis network in eukaryotic cells.** The maintenance of protein homeostasis centers around the turning of gears for multiple cellular processes that oversee the building or breakdown of new or damaged proteins. The four processes of folding, trafficking, synthesis, and degradation rely on pathway systems to maintain the proteome within the cell. For instance, the lysosome–autophagy and ubiquitin–proteasome pathways are central to degradation and recycling of damaged proteins which can accumulate in the cell. On the other hand, ribosome biogenesis aids in the translation and synthesis of new proteins required by the cell. With advancing age, disruption within the proteostasis network can lead to the development of diseases such as sarcopenia and Parkinson’s disease. Created with Biorender.com (accessed on 22 June 2022).

**Figure 2 ijms-23-07602-f002:**
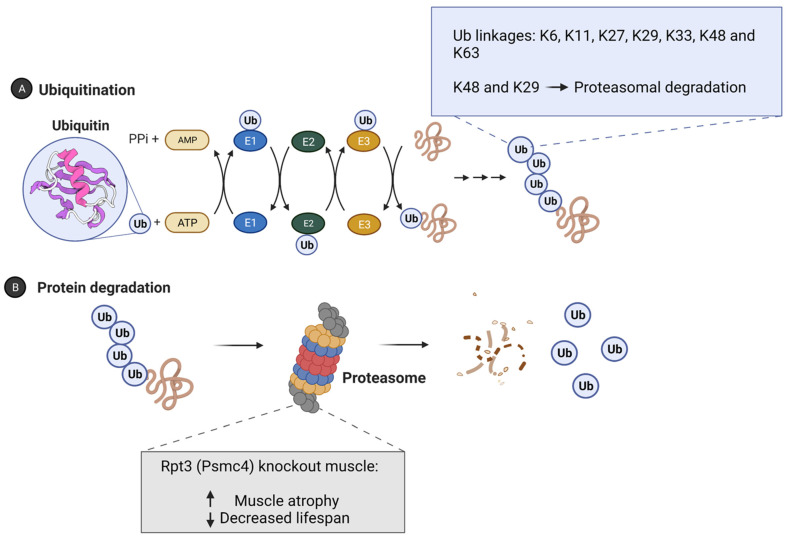
**Overview of the ubiquitin–proteasome system (UPS) and the process of ubiquitination and protein degradation.** The UPS contributes to the proteostasis network by utilizing ubiquitination, a process that involves covalently linking a small protein called ubiquitin to substrate proteins via a series of reactions involving ubiquitin–activating enzymes (E1), ubiquitin–conjugating enzymes (E2), and ubiquitin–protein ligases (E3; ~600 in the human genome (**A**) Ubiquitin, a 76-amino acid protein, is most commonly attached through the formation of an isopeptide bond with a lysine residue on the target protein. A substrate can be monoubiquitinated, multi-ubiquitinated, or polyubiquitinated. The non-proteolytic or proteolytic fate of a substrate protein is determined by the number and/or linkage of ubiquitin chains added. Ubiquitin contains seven lysines (K6, K11, K27, K29, K33, K48, and K63), and it has been suggested that proteins modified with K48 or K29 polyubiquitin-linked chains are targeted to the 26S proteasome for degradation while K63 chains can lead to alterations in signaling and endocytosis (**B**) Rpt3 is an essential subunit of the 26S proteasome core (**B**) and is required for the degradation of substrates selected to the proteasome. In Rpt3 knockout muscles, a significant loss of muscle mass and fiber size, dysregulation of the UPS system and a shortened lifespan has been observed. Created with Biorender.com (accessed on 22 June 2022).

**Figure 3 ijms-23-07602-f003:**
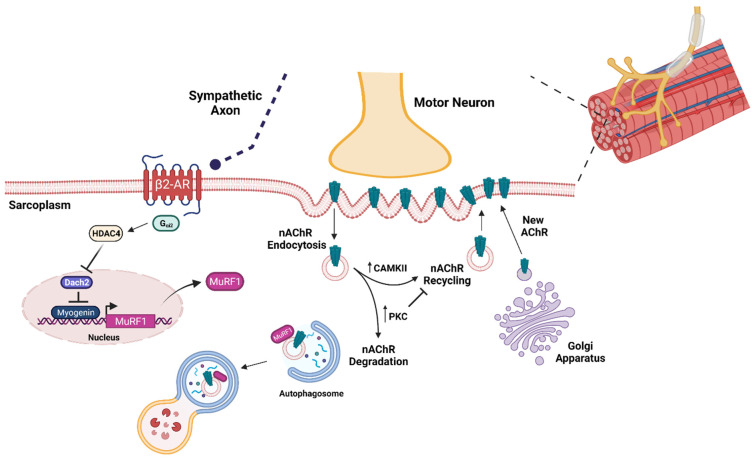
Involvement of the lysosome–autophagosome and ubiquitin–proteasome systems in acetylcholine receptor turnover at the neuromuscular junction. In the Golgi apparatus, newly formed nicotinic acetylcholine rectors (nAChR) are shuttled to the sarcolemma by rapsyn, where under the motor neuron terminal, they cluster together. Turnover of nAChRs begins with endocytosis, and then depending on the signaling pathway activated, the receptors are recycled or degraded. Activation of PKA or CaMKII leads to recycling, while activation of PKC or inhibition of PKA or CAMKII leads to degradation via autophagy. Under atrophy conditions, such as denervation, nAChR degradation is increased and MuRF1 has been shown to colocalize with endocytosed nAChRs and components of the autophagosome. Furthermore, decreases in sympathetic input at the NMJ has been shown to increase nAChR turnover through a HDAC4-myogenin-MuRF1 axis. Created with Biorender.com. Reprinted/adapted with permission from Rudolf and Straka [89] (2022, Wiley and Sons, Hoboken, NJ, USA), and Rodrigues et al. [99] (2022, Elsevier, Amsterdam, The Netherlands).

**Table 2 ijms-23-07602-t002:** Summary of identified E3 ubiquitin ligases involved in aging skeletal muscle to date.

E3 Ligase	Model	Age	Skeletal Muscle	Expression	Reference
MuRF1	Rat (Sprague–Dawley)	24 months	TA	Increased mRNA	Clavel et al. [133]
	Rat (Sprague–Dawley)	30 months	GA	no change	Altun et al. [12]
	Rat (Sprague–Dawley)	30 months	GA	decreased mRNA	Edstrom et al. [143]
	Rat (F344BN)	29 months	TA, SOL	No change	Baehr et al. [124]
	Human	85 ± 1 year	VL	Increased	Raue et al. [147]
	Human	72 ± 8 years	VL	No change	Whitman et al. [138]
MAFbx	Rat (Sprague–Dawley)	24 months	TA	Increased mRNA	Clavel et al. [133]
	Rat (Sprague–Dawley)	30 months	GA	decreased mRNA	Altun et al. [12]
	Rat (Sprague–Dawley)	30 months	GA	decreased mRNA	Edstrom et al. [143]
	Rat (F344BN)	29 months	TA, SOL	No change	Baehr et al. [124]
	Human	85 ± 1 year	VL	Non-significant increase mRNA	Raue et al. [147]
	Human	72 ± 8 years	VL	No change	Whitman et al. [138]
UBR4	Mice	24 months	TA	Increased	Hunt et al. [128]
	Mice	24 months	SOL	no change	Hunt et al. [128]
Parkin	Mice	18 months	QUAD	Increased	Chen et al. [114]
	Mice	24 months	TA	Increased (mitochondrial)	Yeo et al. [102]
	Human	>65 years (Inactive, Frail)	VL	decreased	Drummond et al. [146]
Mib1	Mice	16–30 months	GA	decreased	Seo et al. [129]

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
