# Peer review of "Ubiquitin Ligases in Longevity and Aging Skeletal Muscle"

_ijms, 2022, doi:10.3390/ijms23147602_

Round 1

Reviewer 1 Report

This review paper summarizes recent advances on the role of UPS in skeletal muscle functions and longevity. Overall the manuscript is very well organized and covers most up-to-date information. The underlying molecular mechanisms were also clearly discussed. I believe this review will be of great value for inspiring interventions that prevent aging.

One suggetion I can give is that the role of cullin-ring ubiquitin ligase, the largest family of ubiqutiin ligase, can be discussed in greater details. For example, the cullin-3-RING ubiquitin ligase has been reported to be essential in skeletal muscle functions in mice. Also, the authors may want to briefly discuss the assembly and regulation of CRL ubiquitin ligases to inspire new therapeutics.

Author Response

This review paper summarizes recent advances on the role of UPS in skeletal muscle functions and longevity. Overall the manuscript is very well organized and covers most up-to-date information. The underlying molecular mechanisms were also clearly discussed. I believe this review will be of great value for inspiring interventions that prevent aging. 

We thank the reviewer for their kind comments and for understanding the scope of our discussions within the manuscript. 

One suggetion I can give is that the role of cullin-ring ubiquitin ligase, the largest family of ubiqutiin ligase, can be discussed in greater details. For example, the cullin-3-RING ubiquitin ligase has been reported to be essential in skeletal muscle functions in mice. Also, the authors may want to briefly discuss the assembly and regulation of CRL ubiquitin ligases to inspire new therapeutics.

We thank the reviewer for their suggestions and we have attempted to include details and context around the CRL ubiquitin ligases within the manuscript where we feel appropriate. 

Reviewer 2 Report

Hughes et al. provide a thorough review of the ubiquitin proteasome pathway in muscle, with a focus on the aging muscle where ubiquitin-proteasome mediated protein degradation contributes to sarcopenia, and consequently frailty. 

This field is vast and at present most experimental research in this field tries to connect results on the molecular level such as the expression level of a particular E3 ligase to macroscopic changes such as muscle weight or innervation without clear understanding of the mechanisms that connect the processes on these two very different levels of organization. It is a challenge to review a field at this stage and in my opinion the authors have succeeded at this task, including data from invertebrate model organisms as well as from mice and medically highly relevant human data. The review is improved by their willingness to present contradictory results that have been reported in the literature without the attempt to prematurely rationalize these results. Summaries in the form of tables provided are well organized and useful. The figures are well-made and illustrate the general concepts discussed (note: spelling mistake in figure 2 "decreased").

This review is relevant and well-written and deserves publication when several small issues (see below) have been rectified.

Major issues:

The authors state (68-71) that ligation of ubiquitin chains to proteins changes their conformation and determines whether they are degraded by the proteasome. To my knowledge, this statement does not align with the current understanding of specificity in ubiquitination. As the authors explain correctly in the preceding section, ubiquitin chains can be constructed using each of the seven lysines present on the surface of ubiquitin. Depending on the linkage, the chains adopt starkly different conformations and differ in their biophysical properties, allowing specific binding domains and deubiquitinases to arise. As an example, K48-linked ubiquitin chains adopt a closed, dense conformation while K63-linked chains adopt an extended, open conformation and highly specific binding domains exist for both. Linkage-specific conformation changes of the ubiquitin chains have been extensively researched by e.g. the Komander lab (excellent review: PMID: 22524316). I am not aware of studies showing a conformation change of the ubiquitinated protein, induced by ubiquitination, and dependent on the linkage of the attached ubiquitin chain. I also find it a bit puzzling that none of the three cited papers makes this statement. I would suggest the authors to rephrase the sentence, or, if they indeed wish to make this statement, to include references that report this novel phenomenon. 

261 What are the 9-month old mice compared to?

Minor issues:

138 Species name italics

171 Kitajima

195 Kitajima

294 comma

325 formatting

397 Drosophila (also, should be italic)

561 versus

593 comma

613, 644 remove formatting tags

Author Response

Hughes et al. provide a thorough review of the ubiquitin proteasome pathway in muscle, with a focus on the aging muscle where ubiquitin-proteasome mediated protein degradation contributes to sarcopenia, and consequently frailty. 

This field is vast and at present most experimental research in this field tries to connect results on the molecular level such as the expression level of a particular E3 ligase to macroscopic changes such as muscle weight or innervation without clear understanding of the mechanisms that connect the processes on these two very different levels of organization. It is a challenge to review a field at this stage and in my opinion the authors have succeeded at this task, including data from invertebrate model organisms as well as from mice and medically highly relevant human data. The review is improved by their willingness to present contradictory results that have been reported in the literature without the attempt to prematurely rationalize these results. Summaries in the form of tables provided are well organized and useful. The figures are well-made and illustrate the general concepts discussed (note: spelling mistake in figure 2 "decreased").

We thank the reviewer for their positive feedback and understanding for the scope of our manuscript. It is greatly appreciated by the authors. 

This review is relevant and well-written and deserves publication when several small issues (see below) have been rectified.

Major issues:

The authors state (68-71) that ligation of ubiquitin chains to proteins changes their conformation and determines whether they are degraded by the proteasome. To my knowledge, this statement does not align with the current understanding of specificity in ubiquitination. As the authors explain correctly in the preceding section, ubiquitin chains can be constructed using each of the seven lysines present on the surface of ubiquitin. Depending on the linkage, the chains adopt starkly different conformations and differ in their biophysical properties, allowing specific binding domains and deubiquitinases to arise. As an example, K48-linked ubiquitin chains adopt a closed, dense conformation while K63-linked chains adopt an extended, open conformation and highly specific binding domains exist for both. Linkage-specific conformation changes of the ubiquitin chains have been extensively researched by e.g. the Komander lab (excellent review: PMID: 22524316). I am not aware of studies showing a conformation change of the ubiquitinated protein, induced by ubiquitination, and dependent on the linkage of the attached ubiquitin chain. I also find it a bit puzzling that none of the three cited papers makes this statement. I would suggest the authors to rephrase the sentence, or, if they indeed wish to make this statement, to include references that report this novel phenomenon. 

We have rephrased the sentence highlighted by the reviewer and thank the reviewer for the suggested literature. 

261 What are the 9-month old mice compared to?

We have included this detail in the text. 

Minor issues:

138 Species name italics

171 Kitajima

195 Kitajima

294 comma

325 formatting

397 Drosophila (also, should be italic)

561 versus

593 comma

613, 644 remove formatting tags

We have amended the text to correct all the suggestions made by the reviewer.